# Impact of Derived Features from the Controlled Environment Agriculture Scenarios on Energy Consumption Prediction Model

**Yifan Cao** [1], **Yangda Chen** [1], **Mingwen Shi** [1], **Chuanzhen Li** [1], **Weijun Wu** [1], **Yapeng Li** [1], **Xuxin Guo** [1] and **Xianpeng Sun** [1,2,3,*]

1. College of Horticulture, North West Agriculture and Forestry University, Xianyang 712100, China
2. Key Laboratory od Horticultural Engineering in Northwest Facilities, Ministry of Agriculture, Xianyang 712100, China
3. Facility Agriculture Engineering Technology Research Center of Shaanxi Province, Xianyang 712100, China
* Correspondence: sunxianpeng@nwsuaf.edu.cn

**Abstract:** The high energy consumption CEA building brings challenges to the management of the energy system. An accurate energy consumption prediction model is necessary. Although there are various prediction methods, the prediction method for the particularity of CEA buildings is still a gap. This study proposes some derived features based on the CEA scenarios to improve the accuracy of the model. The study mainly extracts the time series and logical features from the agricultural calendar, the botanical physiological state, building characteristics, and production management. The time series and logical features have the highest increase of 2.8% and 3.6%, respectively. In addition, four automatic feature construction methods are also used to achieve varying degrees of influence from −9% to 8%. Therefore, the multiple feature extraction and feature construction methods proposed in this paper can effectively improve the model performance.

**Keywords:** Controlled Environment Agriculture; energy consumption prediction; derived features; feature engineering





## 1. Introduction

Controlled Environment Agriculture (CEA) is an important method for agricultural production to improve the yield and quality of agricultural products. In CEA, the precise and strict environmental demand leads to high energy consumption and the fluctuation of power load. In order to clarify the energy demand schedule, it is necessary to predict the energy consumption of CEA buildings. This work is also necessary for participating in the dynamic energy market and managing energy production capacity, energy storage, and transformation (Figure 1). At present, this topic is not only an academic discussion, but also a practical problem in society. It is reported that the Shouguang county government and electric power enterprises urgently need this grid technology to stabilize the power network [1].

A highly accurate energy consumption prediction model is the key. In order to obtain a highly accurate energy consumption prediction model, there are two main methods used in the research of building energy consumption prediction, the so-called white-box model and the black-box model. The white-box model is based on dynamic simulation of buildings. The black-box model is a data-driven method based on statistics. The white-box model mainly depends on the dynamic building simulation software such as Energy plus, IDA ICE, and TRANSYS. Energy Plus is software with flexible third-party extensions. It is applicable to a variety of architectural scenarios. It determines the energy load schedule through the dynamic simulation of meteorological, human activities, building physical structure, building materials, and other parameters [2]. The principle of Energy plus includes two key elements, the static model of thermophysical function and the dynamic boundary parameters. Some studies highlighted [3] that, in the hourly comparison between

the measured value and the predicted value of eight buildings in Kentucky. The overall average deviation reached 22% of the standard deviation in one year, and the average deviation of non-heating, ventilation, and air conditioning predictions exceeded 40%. In buildings without heating, ventilation, and air conditioning systems, Energy plus achieved poor performance. For CEA buildings, the model performance of Energy plus cannot be inferred. IDE ICE software has a similar function by inputting the structural parameters of the buildings. The work of the software also needs to combine the embedded or user-edited boundary functions [4]. A study using IDE ICE software revealed the method of optimizing the window–wall ratio using user-edited maintenance structure functions and weather parameters [5]. Ventilation simulation is the strong point of IDA ICE. In previous research, the author compared the two ventilation models through IDA ICE software and found that the ventilation control based on the author's idea can save 30% of energy [6]. In CEA buildings, ventilation is an event that has a great impact on heat load. Therefore, IDE ICE may be applied to energy consumption prediction models. TRANSYS software has been used to study the heat demand of greenhouse buildings. The researchers set different radiation parameters for the model and compared the simulated dynamic temperature field with the actual test. Finally, a set of optimal parameter settings is determined [7]. This study established the optimal parameter settings of the greenhouse model in TRANSYS. The energy consumption has not been simulated. Another study found that user-edited envelope parameters are important for model construction in TRANSYS. Recently, the Computational Fluid Dynamics has also been used to predict building energy consumption. First, the developer uses the model software to build a spatial model. Then, the simulation is carried out according to the target parameters, initial conditions, and boundary conditions. Finally, the developer can obtain the operating power and time of the equipment [8]. In addition, some researchers [9,10] use Computational Fluid Dynamics to simulate the outdoor environment and Energy plus to simulate the indoor environment. It improves overall model performance.

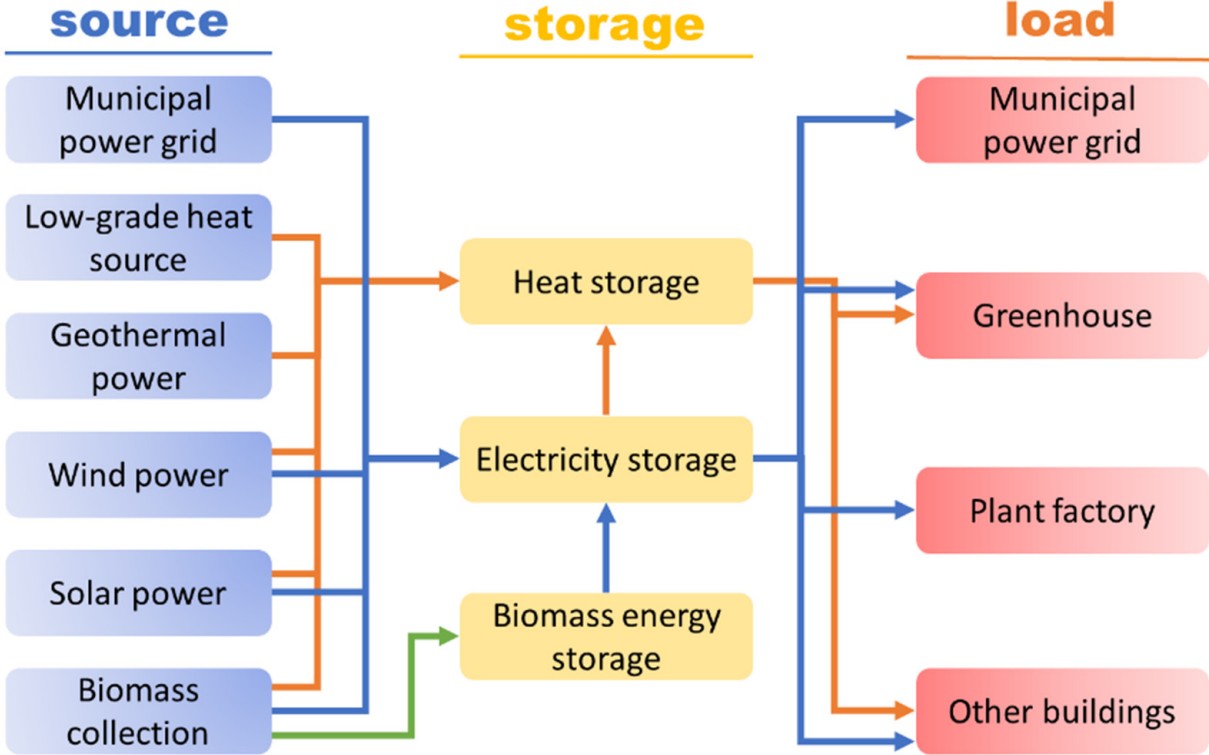

**Figure 1.** The energy network of CEA industrial precinct.

The black-box methods have become popular because of the good computer hardware performance and the maturity of data mining tools. Artificial neural network (ANN) method is one of the earliest models used to predict building energy consumption [11]. ANN algorithms learn the association through data similar to the human nervous system. ANN method is continuously developing and changing. Many new neural network algorithms have been formed [12]. Complex ANN models can be built by changing the number of layers and the location, number, connection mode, and activation function of neurons between layers. Recurrent neural network (RNN) models evolved from ANN models by changing the architecture of the model. the RNN model generates the intermediate variable for storing information in each training and transmits the intermediate variable to the next training. This is the advantage of the RNN model. RNN algorithms were originally used to deal with lexical cohesion in natural language processing. Then, it was found that it has a good performance in dealing with the sequential data. Most prediction problems conform to this characteristic. So, RNN model is often used in prediction problems. It is worth noting that the long short-term memory algorithm evolved from RNN has significantly improved the accuracy of the model due to its processing in feature time series and has become a research hotspot [13–15]. In addition, decision tree model series, such as classification and regression tree and random forest (RF), are also used for comparison in building energy consumption prediction [16]. Support vector machine, autoregressive integrated moving average model, and k-nearest neighbor are common methods [12,17]. The study from Razakolu-Ajayi systematically compared ANN, gradient boosting, deep neural networks, random forest, stacking, k-nearest neighbor, support vector machine, decision tree, and linear regression in predicting the annual energy consumption of buildings. It was found that the deep neural networks model was most suitable for this scenario according to model performance and training cost [18]. In data-driven methods, algorithm structure and model construction are the main research fields at present. For example, in 2016, Jiaoliao Chen et al. [19] used a particle swarm optimization algorithm and genetic algorithm to optimize the hyperparameter optimization process of the greenhouse energy consumption prediction model. In the white box model, people tend to use the self-editing extension to improve the accuracy of the model. For example, Md ShamimAhamed [20] developed a third-party extension in TRANSYS model. They considered insulation blankets and the thermal environment to predict a Chinese solar greenhouse's model performance.

Some scholars have also studied the energy consumption of CEA buildings from the perspective of model predictive control (MPC) (Figure 2, blue part). Ahmed Ouammi and others [21] limited the temperature, light, and water demand of multi-smart greenhouses and predicted the controlled parameters to determine the starting status of the electrical equipment, so as to predict the energy consumption and effectively reduce unnecessary operation. This method is also used for MPC of a semi-closed greenhouse, a CEA building. Guoqing Hu and others [22] proposed a nonlinear model based on a linear model to improve the accuracy of prediction. In another study [23], the Kalman filter algorithm was used to predict the environmental sensor parameters in the greenhouse and, combined with the operating parameters of the equipment, the energy consumption was finally predicted. Interesting research [24] is presented in the case of active late heat storage. Because active late heat storage has strong thermal inertia, prediction is a good way to optimize operation. However, the idea of the above method is to predict the operation status of energy-consuming equipment through environmental and other parameters and, finally, realize energy saving. The other method is to bypass the determination of equipment parameters and predict energy use.

There are few models to directly predict the energy consumption of CEA buildings (Figure 2, orange part). According to the research of Kai Zhang and others [25], the equipment requiring electricity is divided into time-shifting load and non-time-shifting load. Time is used to distinguish equipment energy consumption. This practice does not take into account the impact of climate factors on the building, and the impact of climate factors is significant, especially the operation of fill light and temperature regulation

equipment. MARIO Trejo-Perea and others [26] tried to predict the energy consumption of a 1000-square-meter Venlo type in Mexico by using two environmental features, temperature, and humidity. They also chose the day of the week and the hour of the day as time features and regression as the method. They concluded that, at the 95% confidence level, the performance improvement of the model was significant. P.J.C. Voller-Finck [27] used online meteorological data to predict greenhouse heat load. He found that models with forecasting weather features performed better than those without. In addition, work from FarhatMahmood [28] predicted greenhouse temperature for Model Predictive Control. They added the fan speed and other equipment features of the ventilation system as features to improve the performance of the model. The above methods focus on the research of model algorithms and prediction methods, but few people pay attention to feature engineering. CEA architecture has many of its own particularities that have been ignored. The particularity of CEA building may greatly help to improve the accuracy of the model. Therefore, we decided to investigate the characteristics of CEA building scenarios and verify its effect.

However, for CEA, a special building designed for plants, the derived features from the scene may have a better performance. Feature extraction is a common method to improve model performance. A feature extraction technology using empirical mode decomposition improves the performance of energy consumption prediction models [29]. In addition, three deep-learning-based feature extractions using fully connected autoencoders, evolutionary autoencoders, and generative adaptive networks also significantly improved the performance of the model [30]. In this work, we try to achieve feature extraction from the agricultural calendar, the botanical physiological state, building characteristics, and production management to improve the performance of the CEA building energy consumption prediction model. In addition, we try several automatic feature extraction methods to improve the performance of the model.

In summary, this study was conducted in order to obtain an accurate CEA building energy consumption prediction model to improve energy efficiency and reduce costs. Few studies are focusing on the particularity of CEA buildings to extract features to improve model performance. By comparing previous studies, we propose a method of feature extraction based on CEA scenarios and automatic feature extraction to improve the accuracy and generalization of the model (Figure 3). We think that derived features from the Controlled Environment Agriculture scenarios can improve the model performance.

This article consists of two main parts, as follows:

Verifying the performance of the time series features and logic features based on production management, agricultural calendar, plant physiology, and architectural physics.

Verifying the performance of feature construction methods and summarizing the laws of feature construction methods.

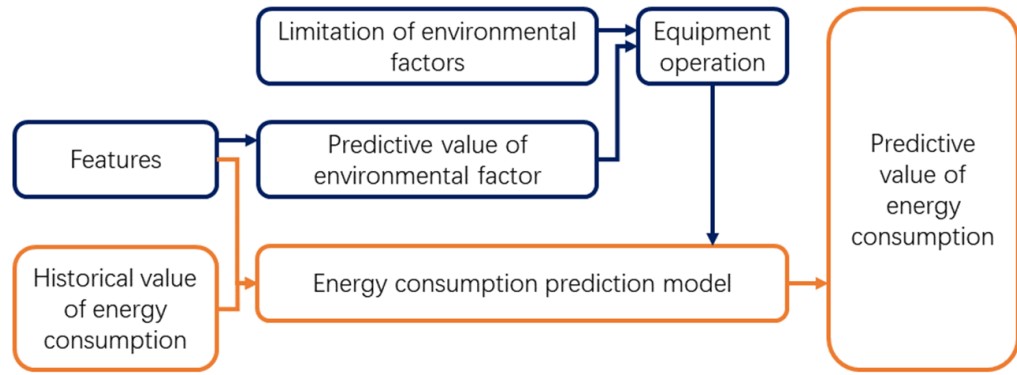

**Figure 2.** Two methods of CEA building energy consumption prediction.

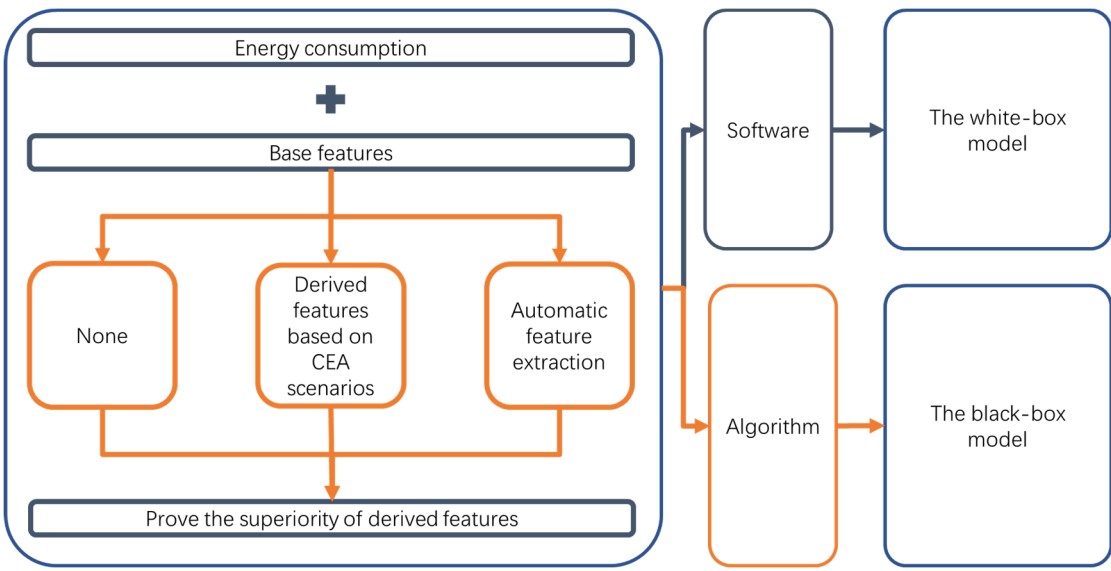

**Figure 3.** Flowchart of this study.

## 2. Methodology

### 2.1. CEA Scenario Particularity Analysis

The CEA buildings are different from residential buildings in the construction and usage habits of shape, envelope, building skin, HVAC system, underlying surface, and equipment [31]. The CEA building covers a large area with a low height (Figure 4a). The atmospheric pressure, light, and thermal environment may be different from the buildings in the city. Building skin material is a significant characteristic of CEA buildings. Except for plant factories, the CEA building has two basic requirements for skin materials (Figure 4b). One is that the material should be able to pass through visible light. This is because visible light is the main source of energy for plant photosynthesis. In addition, the skin material also blocks infrared radiation as much as possible. Greenhouses can reduce the loss of radiant heat by reflecting infrared light from the inside. These characteristics of skin materials directly affect the radiation and thermal environment inside the greenhouse. This has an impact on building energy behavior.

In addition to the differences in architecture, the physiological needs of plants are more significantly different from those of human beings. In production, the main consumption of energy comes from the thermal environment and light environment regulation. In the one-year observation of the test object, 60% of the energy is used for warming and cooling. In addition, 25% of the energy is used for light compensation. Temperature management mainly depends on plant demand and indoor temperature. The indoor temperature is affected by outdoor meteorological factors. The manager's decision on whether to use light compensation depends on five factors, namely plant rhythm, light intensity, temperature, carbon dioxide concentration, and humidity. Plant rhythm is an important physiological activity to maintain plant physiological processes. Any operation in horticultural production will follow this characteristic of plants. Therefore, some features based on plant rhythm may enhance the CEA building energy consumption prediction model. In addition, the plant demand is dynamic (Figure 4c). For example, the tomato greenhouse goes through the seedling stage, vegetative growth stage, flowering and fruiting stage, and fallow stage in a year. In addition, the lettuce greenhouse adopts the recycling production model. Its cultivation density will change in different periods. Managers generally change their management strategies according to the calendar and the growing status (Figure 4d). This directly leads to dynamic changes in energy consumption. This is a difference between CEA buildings and residential buildings.

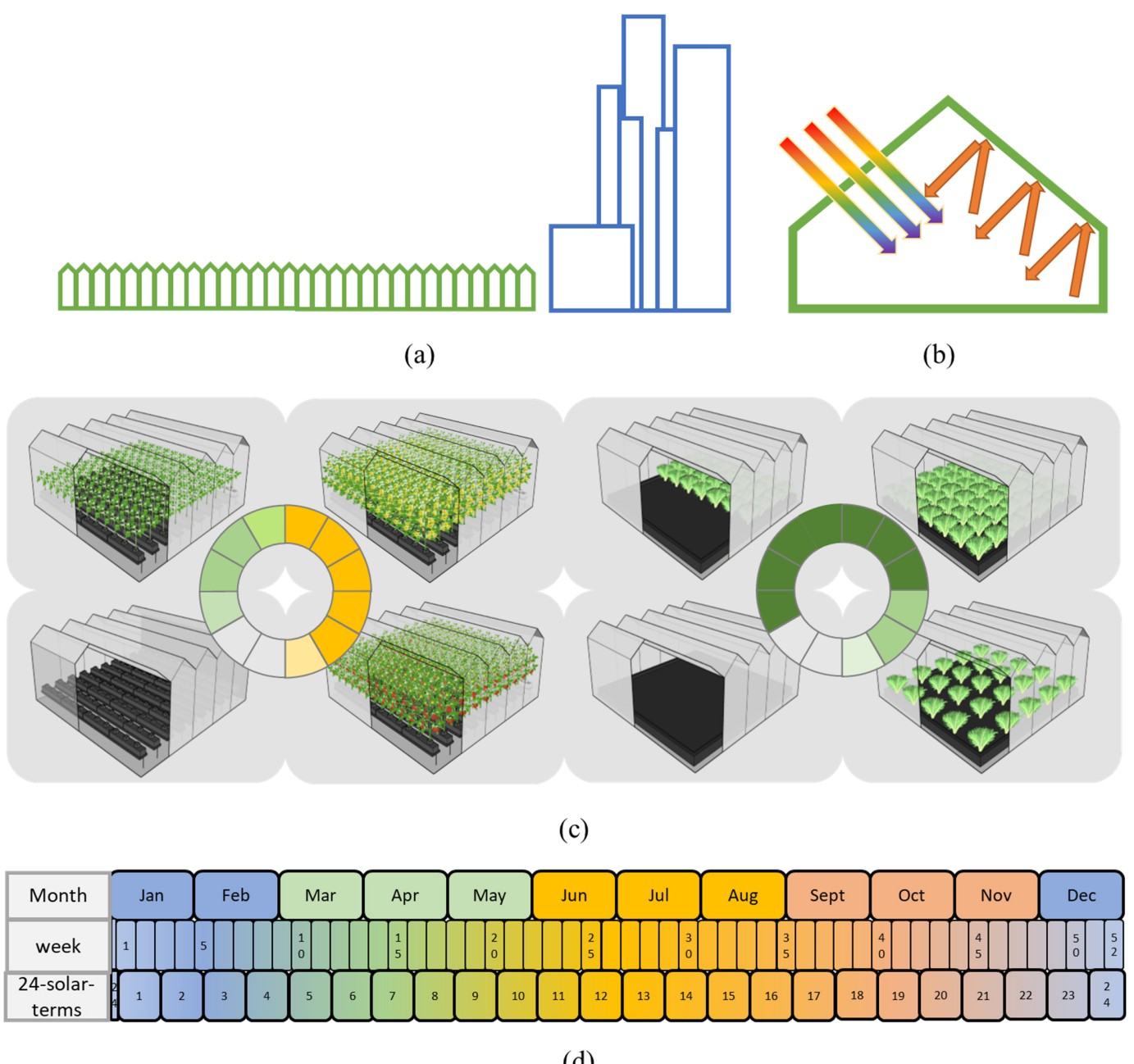

**Figure 4.** Characteristics of CEA buildings. (**a**) The shape and underlying surface of the CEA building area and the city. (**b**) Skin material characteristics. (**c**) The dynamic demand of tomato and lettuce greenhouses. (**d**) Calendar and season.

*2.2. The Basic Situation of the Research Objects*

The CEA buildings are located in Yangling District, Shaanxi Province, China. Three greenhouses are used for production, producing cherry tomatoes, lettuce, and flowers. The building structure details are shown in Table 1. Models: hourly electricity load prediction for cherry tomato and lettuce greenhouses; hourly heat load forecast for these three greenhouses. According to the data, the main electrical load of the greenhouse comes from the supplementary light, while the flower greenhouse was not equipped with supplementary light during the test period. We exclude the flower greenhouse as the target of power load prediction. This is also in line with the needs of practical engineering. The equipment and heating in the greenhouse are shown in Figure 5.

**Table 1.** Structure parameters of CEA buildings.

|  | Flowers Greenhouse | Cherry Tomatoes Greenhouse | Lettuce Greenhouse |
|---|---|---|---|
| Area (m$^2$) | 4180 | 12,744 | 12,744 |
| Gutter height (m) | 5 | 6.5 | 6.5 |
| Ridge height (m) | 7.2 | 7.5 | 7.5 |
| Covering material | Double-layer inflatable PO plastic film | Float glass | Float glass |
| Cultivation mode | Seedbed cultivation | Deep Flow Technique | Substrate culture |

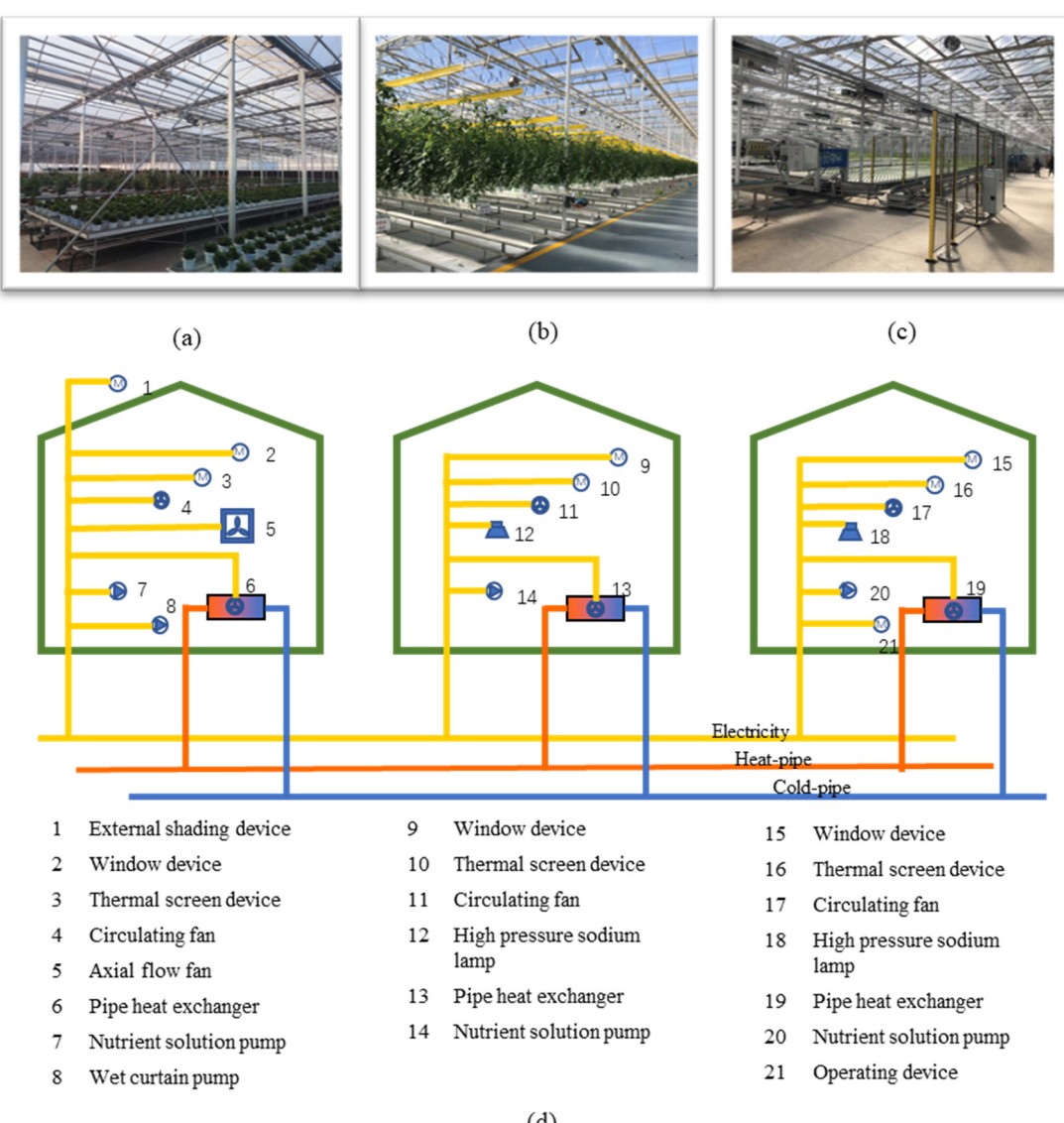

**Figure 5.** The three greenhouses, the heat–cold system, and the electric equipment. (**a**) Flower greenhouse. (**b**) Tomato greenhouse. (**c**) Lettuce greenhouse. (**d**) From left to right are the energy-consuming equipment of flower, lettuce, and tomato greenhouses.

*2.3. Algorithms, Data, and Hardware*

Considering the consumption of computing power and time, the experiment selects the best model from the simple artificial neural network (ANN), classification and regression tree (CART), gradient boosting tree model (GBDT), support vector machine (SVM), linear regression (LNR), logistic regression (LOR), k-nearest neighbor algorithm (KNN), and random forest (RF). The hyperparameter adjustment of each algorithm is shown in Table 2.

**Table 2.** Hyperparameter adjustment.

| Algorithms | Hyperparameter Adjustment |
|---|---|
| ANN | 3 hidden layers; number of hidden layer neurons (30 neurons); learning rate (0.001, 0.01, 0.1, 1) |
| CART | Max depth (from 1 to 25); min samples split (from 2 to 11) |
| GBDT | Estimators (from 50 to 150) |
| SVM | Penalty coefficient (0.1, 1, 10, 100, 1000) |
| LNR | - |
| LOR | Penalty coefficient (0.1, 1, 10, 100, 1000); solver (liblinear, lbfgs, newton-cg, sag); Regularization (l1, l2) |
| KNN | Neighbors (from 2 to 21) |
| RF | Max depth (from 1 to 25); estimators 150 |

The dataset is divided into the training set, validation set, and test set according to the ratio of 3:1:1. The data are generated hourly from 2021.3 to 2022.2. The collected data are 7 kinds of indoor and outdoor temperature, indoor humidity, indoor carbon dioxide concentration, outdoor light intensity, wind speed, and wind direction. Software: Python3.8.5, scikit-learn 0.24.2; hardware: Elastic Compute Service from Alibaba Cloud Computing Co., Ltd. (32 vCPU AMD EPYC™ ROME 7H12, RAM 64GiB) and PC (Intel(R) Core(TM) i5-10210U CPU, RAM 8 GiB).

*2.4. The Experimental Treatment and Its Principle*

The experiment is divided into 7 items, including verifying the effect of two types of features, namely time series features and logic features, and studying the effect of five construction features, namely single-heat coding feature, exponential feature, equal-frequency binding feature, K-means clustering binding inheritance feature, double-feature polynomial coupling feature, and double-feature cross-combination feature. The processing principle is as follows. See Table 3 and Figure 6 for the processing settings.

**Table 3.** Treatments setting.

| Features and Treatment | Processing Settings | Code |
|---|---|---|
| basic features | Indoor and outdoor temperature, indoor humidity, indoor carbon dioxide concentration, outdoor light intensity, wind speed, wind direction | TS0 |
| time series features | In the experiment, the basic feature was used as a blank control, and 9 different time series features were added to the basic feature. | TS1-9 |
| logic features | Taking the basic feature as a blank control, 8 different business-logic-derived features are added to the basic feature, respectively. | LG1-8 |
| one-hot encoding treatment | One-hot encoding is performed on the wind direction, the 24 solar terms, the lunar calendar, the 24 solar terms considering the influence of the moon and the earth, the week, the month, and the hour, and the corresponding results without this processing are selected as the control. | OHE1-7 |
| equal frequency binning treatment | For outdoor temperature, indoor temperature, indoor humidity, indoor carbon dioxide concentration, outdoor light intensity, wind speed, outdoor temperature change rate, indoor temperature change rate, indoor and outdoor temperature difference, indoor humidity change rate, indoor temperature and humidity product, indoor carbon dioxide concentration change rate, and the outdoor illumination change rate were divided into 4, 5, and 6 boxes by equal frequency binning treatment, and the corresponding results without this treatment were selected as the control. | 4EFD1-13 5EFD1-13 6EFD1-13 |
| K-means cluster binning treatment | The same as equal frequency binning treatment but it uses K-means cluster binning | 4KMD1 5KMD2 6KMD3 |
| polynomial encoding treatment | A total of 24 features, including basic features, time series features, and business-logic-derived features, are subjected to dual-feature polynomial coupling and change rate processing, and 576 groups of processing groups are obtained for comparison with the above-mentioned 24 untreated groups. | MPN1-576 |
| cross-combination treatment | The cross-combination feature construction feature selects two different time series, that is, the daily cycle and the annual cycle, for cross-combination, and 18 groups of treatment groups are obtained as a comparison with the corresponding results without this treatment. | BCC1-18 |

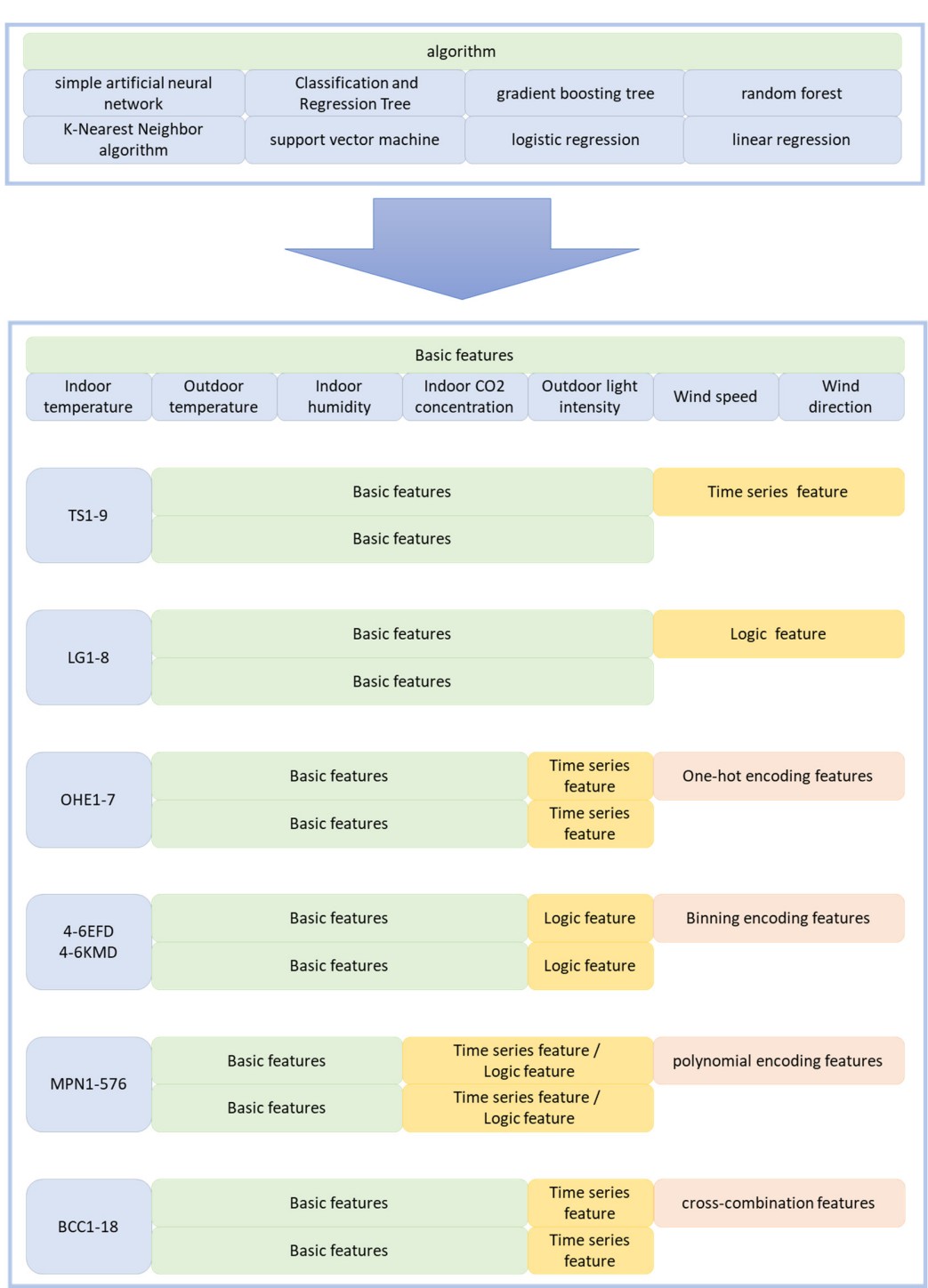

**Figure 6.** Experiment settings.

2.4.1. Time Series Features

Time series features include the 24 solar terms, the 24 solar terms considering the influence of the moon and the Earth, the Chinese lunar calendar, fallow days, the weeks of a year, months, day–night markers, the photosynthetic state, and the hours of the day.

The feature named "moon-24-solar-terms" is derived from the research results of Zhong [32]. The work mentions that the influence of lunar gravity on the Earth's atmospheric motion and tides can improve features of the 24 solar terms. The "24-solar-terms" and the "lunar calendar" are generally recognized in Chinese society [33]. There are two reasons for choosing calendars such as months, weeks, the 24 solar terms, and others as

time series features. People have summarized the calendar through astronomy, meteorology, and phenology [34]. These factors are strongly related to the objective conditions of agricultural production. On the other hand, calendars play a guiding role in people's life and production schedule, especially agricultural production schedules [35]. In a few words, the time series from the calendar contains information on both the impact of the Earth's revolution and rotation on nature and the inspiration for human activities.

The "fallow day" and the "photosynthetic state" are mainly from the work schedule submitted by the project manager. The "fallow day" refers to whether the greenhouse is in a production state or a rest state. The "photosynthetic state" is an information tag of the plant's photosynthesis state. Plants have their own physiological rhythms. The production manager will also follow this rule to adjust the compensation light. See Table 4 for specific processing and codes.

**Table 4.** Time series feature treatments.

| Features Code | Features | Control Group |
| --- | --- | --- |
| TS0 | base | - |
| TS1 | base\24-Solar-terms | TS0 |
| TS2 | base\lunar calendar | TS0 |
| TS3 | base\moon-24-Solar-terms | TS0 |
| TS4 | base\fallow day | TS0 |
| TS5 | base\week | TS0 |
| TS6 | base\month | TS0 |
| TS7 | base\day and night | TS0 |
| TS8 | base\photosynthetic state | TS0 |
| TS9 | base\hour | TS0 |

2.4.2. Logic Features

The original features may not achieve the best results. Logic features are based on the logic of equipment operation in management and physical laws. Logic features include the hourly rate of change of outdoor light intensity, indoor and outdoor temperature, indoor humidity, and carbon dioxide concentration; indoor and outdoor temperature difference; indoor temperature and humidity product; and wind speed and direction product. We know that the greater the temperature difference is, the faster the heat transfer speed is. Therefore, the indoor and outdoor temperature difference can be added as a new feature. In addition, the rate of change of features reflects the trend of change. The product of wind speed and direction can distinguish the wind. The product of temperature and humidity is the standard for judging window openings in production.

The rate-of-change treatments for temperature, humidity, and light intensity indicate trends. Using the temperature difference between indoors and outdoors can reveal the direction of heat transfer between the greenhouse building and the outside. The product of temperature and humidity can distinguish the high-temperature and high-humidity state from other states. The product of wind speed and direction makes it vectorized. See Table 5 for specific processing and codes.

**Table 5.** Logic feature treatments.

| Feature Code | Features | Control Group |
| --- | --- | --- |
| LG1 | base\OT change rate | TS0 |
| LG2 | base\IT change rate | TS0 |
| LG3 | base\OT-IT | TS0 |
| LG4 | base\IH change rate | TS0 |
| LG5 | base\IT*IH | TS0 |
| LG6 | base\CO2 change rate | TS0 |
| LG7 | base\OLI change rate | TS0 |
| LG8 | base\WD*WS | TS0 |

### 2.4.3. One-Hot Encoding Feature

One-hot encoding is a common method to expand the number of data features. It converts each value level of the discrete feature with multiple value levels into many individual features, for example, changing the feature of the month into "whether it is January", "whether it is February", and so on. This process will sparse the information but, at the same time, it can increase focus on key value levels. It helps the algorithm to reasonably allocate weights on key values. In order to enable the computer to read the classification features, the traditional practice is to convert the classification features into integers or other numerical values. Seger Cedric [36] believes that this transformation feature will mislead distance and linear algorithms because integers have size and some classification objects may not have been ordered. One-hot encoding turns numbers into Booleans. In this way, in distance or linear algorithms, several dimensions are added instead of several values of the same dimension. In the work of Kim and Kwang Ho, the building energy consumption model using the DNN algorithm achieved an RMSE 41% lower than the method without one-hot encoding [37]. See Table 6 for specific processing and codes.

**Table 6.** One-hot encoding feature treatments.

| Feature Code | Features | Control Group |
| --- | --- | --- |
| OHE1 | base\WD (one-hot) | TS0 |
| OHE2 | base\24-Solar-terms\24-Solar-terms (one-hot) | TS1 |
| OHE3 | base\lunar calendar\lunar calendar (one-hot) | TS2 |
| OHE4 | base\moon-24-Solar-terms\moon-24-Solar-terms (one-hot) | TS3 |
| OHE5 | base\week\week (one-hot) | TS4 |
| OHE6 | base\month\month (one-hot) | TS5 |
| OHE7 | base\hour\hour (one-hot) | TS9 |

### 2.4.4. Equal-Frequency and K-Means Cluster Binning

Equal-frequency binning and K-means binning processing is to bin discrete variables according to equal frequency and K-means clustering. In K-means clustering, the sample is divided into k bins according to variables, so that the sum of Euclidean distances of all variables is the minimum. Binning processing can reduce the interference of variables with a large order of magnitude difference in algorithm learning. In the distance algorithm, a number replacing the approximate values can reduce fluctuations to improve model performance [38]. See Table 7 for specific processing and codes.

**Table 7.** Binning feature treatments.

| Feature Code | Features | Control Group |
| --- | --- | --- |
| 4-6EFD1 | base\OT (EFD) | TS0 |
| 4-6EFD2 | base\IT (EFD) | TS0 |
| 4-6EFD3 | base\IH (EFD) | TS0 |
| 4-6EFD4 | base\CO2 (EFD) | TS0 |
| 4-6EFD5 | base\OLI (EFD) | TS0 |
| 4-6EFD6 | base\WS (EFD) | TS0 |
| 4-6EFD7 | base\OT change rate\OT change rate (EFD) | LG1 |
| 4-6EFD8 | base\IT change rate\IT change rate (EFD) | LG2 |
| 4-6EFD9 | base\OT-IT\OT-IT (EFD) | LG3 |
| 4-6EFD10 | base\IH change rate\IH change rate (EFD) | LG4 |
| 4-6EFD11 | base\IT*IH\IT*IH (EFD) | LG5 |
| 4-6EFD12 | base\CO2 change rate\CO2 change rate (EFD) | LG6 |
| 4-6EFD13 | base\OLI change rate\OLI change rate (EFD) | LG7 |
| 4-6KMD1 | base\OT (KMD) | TS0 |
| 4-6KMD2 | base\IT (KMD) | TS0 |
| 4-6KMD3 | base\IH (KMD) | TS0 |

**Table 7.** *Cont.*

| Feature Code | Features | Control Group |
|---|---|---|
| 4-6KMD4 | base\CO2 (KMD) | TS0 |
| 4-6KMD5 | base\OLI (KMD) | TS0 |
| 4-6KMD6 | base\WS (KMD) | TS0 |
| 4-6KMD7 | base\OT change rate\OT change rate (KMD) | LG1 |
| 4-6KMD8 | base\IT change rate\IT change rate (KMD) | LG2 |
| 4-6KMD9 | base\OT-IT\OT-IT (KMD) | LG3 |
| 4-6KMD10 | base\IH change rate\IH change rate (KMD) | LG4 |
| 4-6KMD11 | base\IT*IH\IT*IH (KMD) | LG5 |
| 4-6KMD12 | base\CO2 change rate\CO2 change rate (KMD) | LG6 |
| 4-6KMD13 | base\OLI change rate\OLI change rate (KMD) | LG7 |

2.4.5. Polynomial Product

The polynomial product is a method to obtain new variables by multiplying or powering existing variables. Table 8 shows the process of order 2, 3, and 4 transformations of two original features. All non-auto-constructing features are processed in this way.

**Table 8.** Polynomial product feature treatment.

| Initial Features | Order | Transformation Features |
|---|---|---|
| $[X_1, X_2]$ | 2 | $[X_1^2, X_2^2, X_1 * X_2]$ |
| | 3 | $[X_1^3, X_2^3, X_1^2 * X_2, X_2^2 * X_1]$ |
| | 4 | $[X_1^4, X_2^4, X_1^3 * X_2, X_2^3 * X_1, X_1^2 * X_2^2]$ |

2.4.6. Cross-Combination

The cross-combination processing includes the following steps. The first two features are separately one-hot coded. Second, each sub feature in the two feature combinations is mu;tiplied. Finally, all the multiplied variables form a new feature combination. A new feature value of "1" means that a certain value level of two original features is met at the same time. For example, in the two original features of the month and day–night markers, there are a total of 24 new Boolean features named "January and day", "February and day", ... "December and day", "January and night", "February and night", ... "December and night".

*2.5. Evaluation*

The paper mainly uses the coefficient of determination as the validation criteria, and the calculation formula is as follows:

$$R^2 = \frac{\sum_{i=1}^{n}(\hat{y}_i - \overline{y})^2}{\sum_{i=1}^{n}(y_i - \hat{y}_i)^2}$$

$y_i$ is the actual value of the test set;
$\hat{y}_i$ is the predicted value;
$\overline{y}$ is the average of the whole dataset.

**3. Results and Discussion**

*3.1. Performance of Eight Algorithms in the Training Model*

The quality of the algorithm depends on the accuracy and generalization of the algorithms. Figure 7 shows the accuracy of the performance of the eight algorithms in the five models. Enhanced tree models such as RF and GBDT perform best in model accuracy. The performance of KNN and SVM algorithms based on Euclidean distance are second only to the enhanced tree algorithm. The accuracy of CART algorithm is better than that of general regression models. In addition, the ANN model set in the test conditions performs

poorly. However, the poor performance of ANN model in the test may be due to the simple structure. For ANN model, only the design of three layers of perceptron may not achieve the best performance. The complex structure of ANN model has always been a hotspot in the research of energy consumption prediction. Through research, it is found that the multi-level LSTM algorithm with periodicity and recency consideration can achieve better accuracy compared with ARIMA, ARFIMA, and BPNN in energy consumption prediction of HAVC systems in buildings [13]. Generalization represents the general performance of algorithms in different loads and buildings. Variance is an indicator used to compare the generalization of algorithms between models. RF has the strongest generalization. In addition, the variance of RF is close to half of the second one (Table 9). The variances of GBDT, KNN, CART, and ANN belong to the same order of magnitude. The variances of the regression algorithms are an order of magnitude higher than the first five places. The generalization of SVM is worse. Among the five algorithms using SVM, the $R^2$ of some models is even less than 0.2. This usually means that the model is invalid.

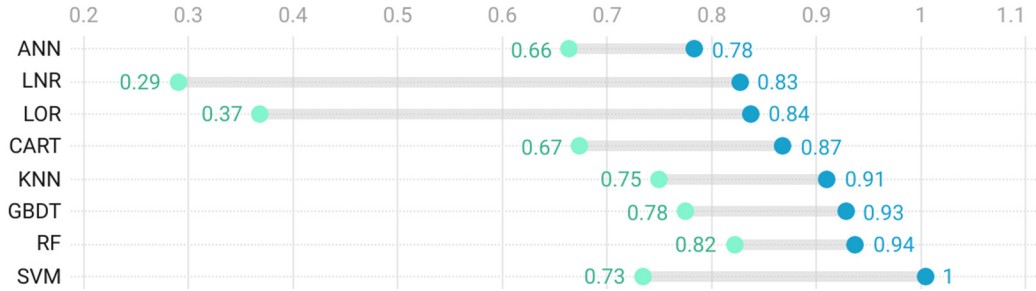

The SVM algorithm does not converge on some models. The lowest R² for SVM is around 0.2.

**Figure 7.** $R^2$ of eight algorithms.

**Table 9.** Variances of eight algorithms.

| Algorithm | Variance |
|:---:|:---:|
| ANN | 0.006902 |
| LNR | 0.058159 |
| LOLI | 0.042397 |
| CART | 0.009389 |
| KNN | 0.005112 |
| GBDT | 0.004605 |
| RF | 0.002635 |
| SVM | 1.814549 |

In summary, the order of algorithm accuracy is enhancement tree algorithm, Euclidean distance algorithm, regression tree algorithm, the regression algorithm, and ANN with simple structure. The order of generalization is RF, other tree algorithms, KNN, ANN with simple structure, the regression algorithm, and SVM. RF is the best choice for accuracy and generalization.

*3.2. Important Features in the Model*

Key features can help developers analyze important influencing factors. The important parameter of the random forest is a good indicator. If a feature can reduce more cross-entropy, the feature is more important. This is the basic principle of this parameter. However, this does not mean that features with lower scores are meaningless. For developing a model, any information gained can help improve the accuracy of the model. However, too many low correlation features may also affect the generalization.

As shown in Figure 8, outdoor temperature and outdoor light intensity are two key features in electric load prediction. The outdoor temperature is the key feature in heat load

prediction. The energy consumption of electric systems mainly comes from lights, which has a high correlation with outdoor light intensity and temperature. In addition, logically, the outdoor temperature is the most critical indicator of heat load. In other words, the effect of the features depends on the energy use behavior of CEA buildings. Therefore, when considering the development of a CEA building energy consumption model, it is necessary to fully evaluate the features of working logic in energy use behavior. To be specific, the fill light system and automatic transportation system may be the cause of high energy consumption. In addition, the differences in energy use behavior of different CEA buildings, regions, and crops also need to be observed. There are two ways to help you find high-energy-consuming electrical equipment. The first way is to install electricity meters for different equipment, then observe the power consumption for a period of time. Another way is to estimate the electric energy according to the rated power and operation time of the equipment. After confirming the high-energy-consumption object, you can surmise the possible key features according to its working logic.

| | Electric load prediction(tomato) | Electric load prediction(lettuce) | Heat load prediction(tomato) | Heat load prediction(lettuce) | Heat load prediction(flower) |
|---|---|---|---|---|---|
| OT change rate | 0.013 | 0.027 | 0.004 | 0.007 | 0.006 |
| OT | 0.191 | 0.220 | 0.868 | 0.653 | 0.752 |
| IT change rate | 0.042 | 0.027 | 0.012 | 0.022 | 0.013 |
| IT | 0.028 | 0.017 | 0.006 | 0.007 | 0.008 |
| OT-IT | 0.039 | 0.046 | 0.007 | 0.027 | 0.020 |
| IH change rate | 0.013 | 0.013 | 0.005 | 0.011 | 0.008 |
| IH | 0.017 | 0.014 | 0.005 | 0.026 | 0.014 |
| IT*IH | 0.035 | 0.016 | 0.005 | 0.006 | 0.016 |
| CO2 change rate | 0.334 | 0.033 | 0.004 | 0.008 | 0.004 |
| CO2 | 0.062 | 0.018 | 0.009 | 0.011 | 0.008 |
| OLI change rate | 0.018 | 0.029 | 0.004 | 0.011 | 0.005 |
| OLI | 0.080 | 0.287 | 0.016 | 0.150 | 0.096 |
| WS | 0.012 | 0.009 | 0.002 | 0.005 | 0.002 |
| WD(raw) | 0.007 | 0.008 | 0.005 | 0.003 | 0.003 |
| WD | 0.002 | 0.003 | 0.001 | 0.001 | 0.002 |
| WD*WS | 0.005 | 0.006 | 0.002 | 0.003 | 0.002 |
| 24-Solar-terms | 0.017 | 0.029 | 0.004 | 0.014 | 0.009 |
| lunar calendar | 0.006 | 0.019 | 0.005 | 0.012 | 0.003 |
| day&night | 0.000 | 0.001 | 0.000 | 0.000 | 0.000 |
| moon-24-Solar-terms | 0.009 | 0.010 | 0.021 | 0.006 | 0.002 |
| fallow day | 0.001 | 0.000 | NaN | 0.001 | 0.000 |
| photosynthetic state | 0.001 | 0.037 | 0.000 | 0.000 | 0.001 |
| week | 0.034 | 0.049 | 0.009 | 0.007 | 0.009 |
| month | 0.003 | 0.009 | 0.003 | 0.003 | 0.004 |
| hour | 0.034 | 0.074 | 0.005 | 0.006 | 0.014 |

**Figure 8.** The importance of features (using RF).

In addition, there is a strong correlation between the prediction of power load and the change rate of carbon dioxide concentration in tomato greenhouse. This may be related to the stronger carbon adsorption capacity of tomato plants. The carbon dioxide adsorption rate of lettuce and tomato is different under the condition of light supplement. Under the condition of light saturation, the carbon dioxide absorption rate of tomato was 37.14 $\mu mol\cdot s^{-1}\cdot dm^{-2}\cdot h^{-1}$[39], while that of lettuce was 20.22 $\mu mol\cdot s^{-1}\cdot dm^{-2}\cdot h^{-1}$ [40]. The leaf area of tomato plants in a three-dimensional state is also larger than that of lettuce. The carbon dioxide absorption rate of tomato is faster than that of lettuce in a greenhouse. As shown in Figure 9, the carbon dioxide concentration in the tomato greenhouse changes more violently. In addition, the carbon dioxide concentration in the tomato greenhouse is always lower than that in the lettuce greenhouse. Light and carbon dioxide are energy sources and raw materials for plant photosynthesis, respectively. Therefore, there is a high correlation between the fill light system and carbon dioxide. In the development of models,

the impact of plant species and varieties on the energy consumption of the fill light system should be evaluated.

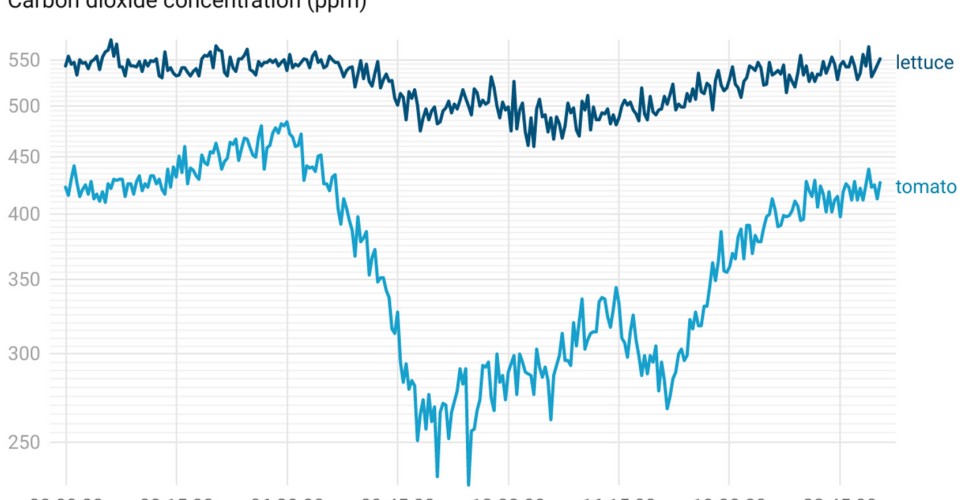

*The figure shows the change of carbon dioxide concentration in tomato and lettuce greenhouses on February 2, 2021.*

**Figure 9.** The change in carbon dioxide concentration in two greenhouses.

### *3.3. Performance of Time Series and Logical Features*

In the annual cycle time series features, the time series based on the Chinese local calendar performs well (Figure 10). Moreover, "24-solar-terms", "moon-24-solar-terms", and "lunar calendar" improved the average performance of all models. Among them, the 24 solar terms improved the average performance of the model by nearly 3%. The improvement of the lunar calendar also exceeds that of the month. It is worth noting that, in the heat load prediction model, the improvement effect of 24 solar terms even exceeds that of weeks (Table 10). This shows that calendar information, especially local calendar information, is worthy of attention for CEA building energy consumption model training. The CEA buildings are engaged in horticultural production. This means that its energy use behavior depends on meteorological, phenological, and other conditions. The weather and phenology are related to the agricultural calendar. Therefore, in the actual project, it is necessary to understand the calendar tools in the local culture, especially the agricultural calendar. In the experiment, the 24 solar terms and lunar calendar originated from farming activities in China. In other projects, developers should pay more attention to the local calendar.

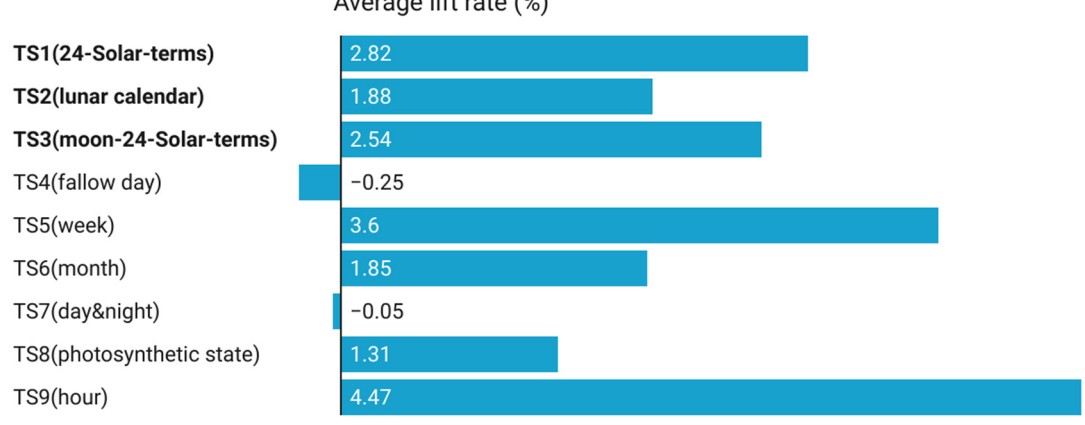

**Figure 10.** Average increase in the $R^2$ of the model (time series features).

**Table 10.** The $R^2$ of each model (time series features).

|  | Electric Load Prediction (Tomato) | Electric Load Prediction (Lettuce) | Heat Load Prediction (Tomato) | Heat Load Prediction (Lettuce) | Heat Load Prediction (Flower) |
|---|---|---|---|---|---|
| TS0 | 0.791647 | 0.667267 | 0.90634 | 0.908397 | 0.878356 |
| TS1 | 0.808981 | 0.714331 | 0.921685 | 0.92328 | **0.891697** |
| TS2 | 0.799595 | 0.698115 | 0.919089 | 0.921532 | 0.886441 |
| TS3 | 0.802259 | 0.709967 | 0.927091 | **0.924127** | 0.886599 |
| TS4 | 0.786413 | 0.661225 | 0.907121 | 0.910077 | 0.878705 |
| TS5 | **0.81279** | **0.733286** | **0.930125** | 0.923851 | 0.888203 |
| TS6 | 0.797521 | 0.698015 | 0.923338 | 0.923517 | 0.881523 |
| TS7 | 0.785515 | 0.669379 | 0.907642 | 0.909785 | 0.877667 |
| TS8 | 0.807516 | 0.694927 | 0.907315 | 0.909097 | 0.880386 |
| TS9 | **0.855509** | **0.750499** | **0.910408** | **0.914261** | **0.884554** |

In the daily cycle time series features, the use of day and night binary markers alone cannot effectively improve the performance of the model. However, the binary marker based on the plant photosynthesis state improved the model performance by more than 1.3%. In CEA buildings, plants usually receive the first ray of light earlier than sunrise. This is the preparation of producers for higher output and quality. The high energy consumption fill light is related to this operation. Therefore, the author divides time into two situations according to whether plants are in the state of photosynthesis. This feature comes from the observation of CEA building production process. In addition, the hour is a very effective feature to improve the effectiveness of the model.

The logic features also have some good features (Table 11). In the change rate feature, indoor and outdoor temperature and outdoor light intensity have improved the performance of the model (Figure 11). The change rate of indoor temperature and outdoor light intensity has improved the electric load prediction model by more than 3.5%. The change rate of carbon dioxide concentration has improved the power load model by more than 0.6%. The change rate of indoor humidity also improved the thermal load model by more than 0.6%. Carbon dioxide is related to the operation of light compensation. Humidity is related to window opening management, and window opening is a common form of heat loss. Surprisingly, the indoor and outdoor temperature difference improves the electric load model more than the heat load model. This is inconsistent with the original assumption. The other two features did not achieve the expected effect. In this paper, the author constructs several new features based on on-site observation and his/her own knowledge. Some features come from light compensation operation and window opening management in horticultural production. Some features come from the actual building thermophysical process. The effect of a single feature may only be consistent with this case, but this observation and artificial excavation method are what the author wants to emphasize.

**Table 11.** The $R^2$ of each model (logic features).

|  | Electric Load Prediction (Tomato) | Electric Load Prediction (Lettuce) | Heat Load Prediction (Tomato) | Heat Load Prediction (Lettuce) | Heat Load Prediction (Flower) |
|---|---|---|---|---|---|
| TS0 | 0.791647 | 0.667267 | 0.90634 | 0.908397 | 0.878356 |
| LG1 | 0.803024 | 0.682579 | 0.9063 | 0.913625 | 0.87894 |
| LG2 | 0.837224 | 0.677027 | 0.911419 | 0.923153 | 0.89338 |
| LG3 | 0.789475 | 0.677239 | 0.906165 | 0.910306 | 0.877164 |
| LG4 | 0.780316 | 0.668154 | 0.907328 | 0.914761 | 0.887967 |
| LG5 | 0.782213 | 0.667545 | 0.908385 | 0.906885 | 0.878493 |
| LG6 | 0.817478 | 0.673809 | 0.906936 | 0.908924 | 0.876753 |
| LG7 | 0.822927 | 0.688554 | 0.91037 | 0.915501 | 0.881089 |
| LG8 | 0.788559 | 0.664845 | 0.905989 | 0.907066 | 0.87949 |

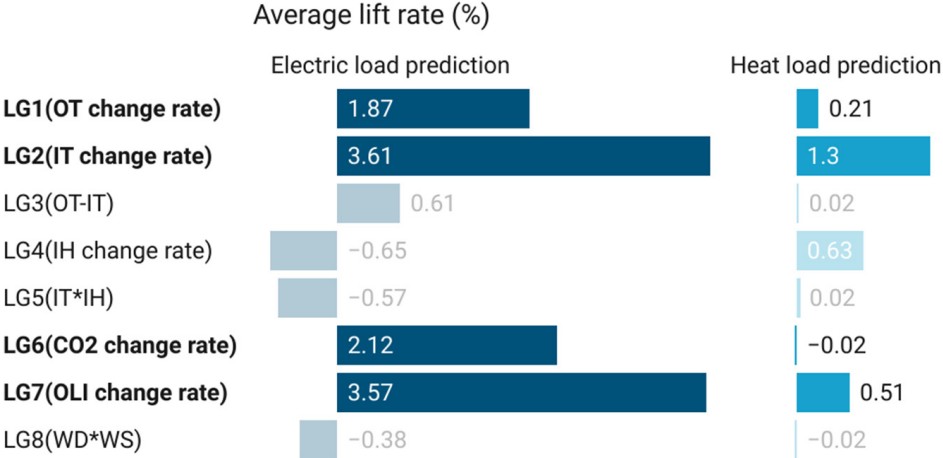

**Figure 11.** Average increase in the $R^2$ of the model (logic features).

*3.4. Automatic Feature Construction Effect*

Some automatically constructing features can also enhance the model's performance. In most cases, the one-hot encoding can improve the performance of the models (Figure 12). Especially for the electric load model, most features of the one-hot encoding process can improve the model by 1.6%~2.0%. The treatment of "week" makes the improvement of the electric load model and heat load model even reach 7.8% and 2.2%. However, the treatment of "month" has a negative effect.

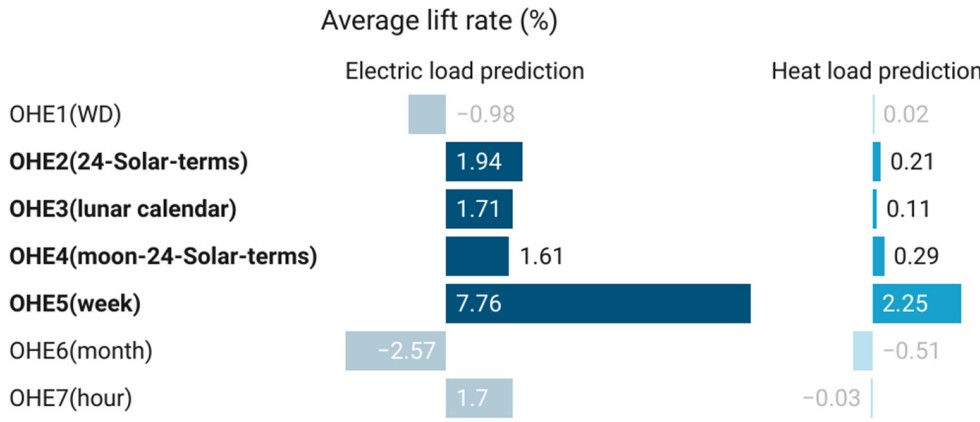

**Figure 12.** Average increase in the $R^2$ of the model (one-hot encoding).

In the binning encoding processing (Figure 13), the influence of original features on the model is greater than that of processing. "The outdoor light intensity" feature performs better than other features. EFD13 and KMD13 achieved a maximum increase of 2.2% and 5.0%, respectively. In addition, the treatment of "the product of temperature and humidity" (EFD11 and KMD11) has a negative impact on the model accuracy. Compared with the equal frequency binning encoding, the effect of the K-means binning encoding treatment on the models is more unstable. The K-means binning encoding processing can achieve good performance in some models. If there is a selection of features, the K-means binning encoding processing should be given priority.

In the polynomial encoding processing (Figure 14), treatments have little impact on the models. The overall improvement did not exceed 0.6%. The "week" and "carbon dioxide concentration" features have a stable and good improvement effect. Another obvious characteristic is that most of the time series features can be improved. In the case of limited computing resources, this method can be abandoned preferentially.

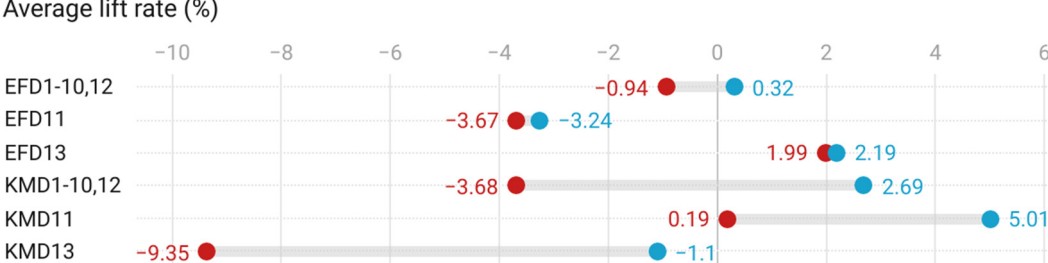

**Figure 13.** Average increase in the R$^2$ of the model (binning encoding).

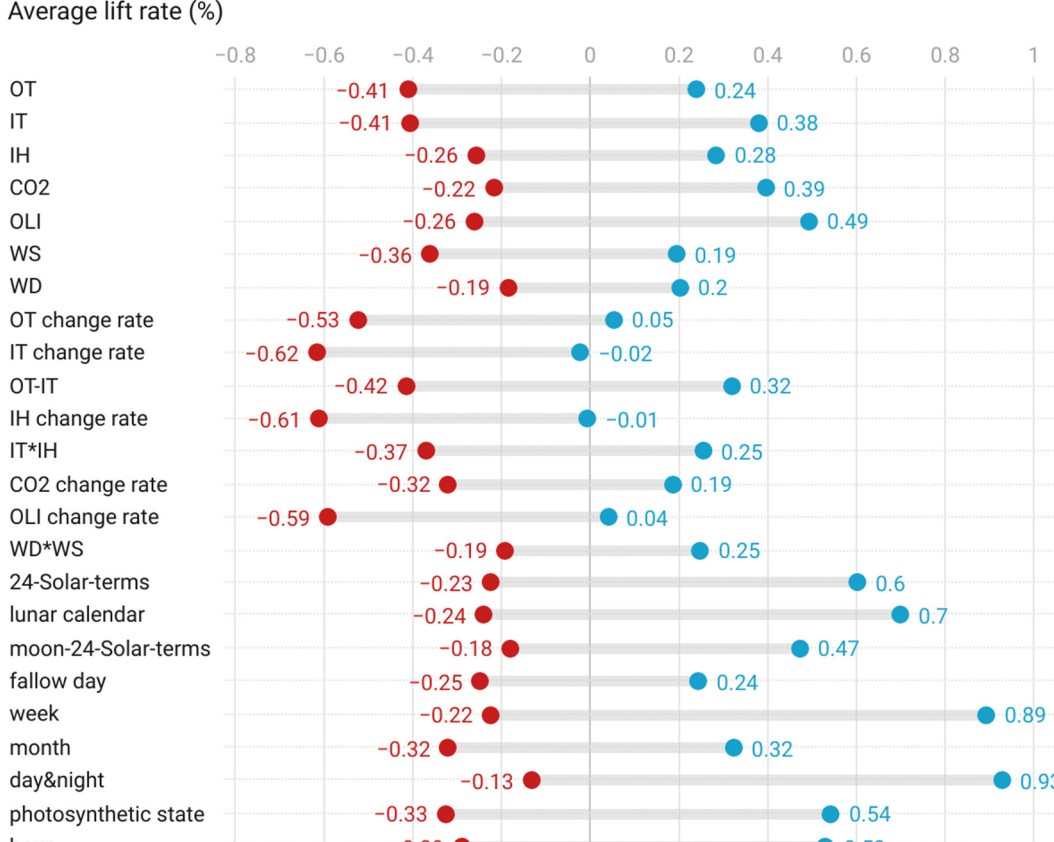

**Figure 14.** Average increase in the R$^2$ of the model (cross-combination encoding).

The cross-combination encoding processing has little improvement on the model (Figure 15) but the improvement is relatively stable. The maximum increase in the model does not exceed 0.8%. The "day & night", "plant states", "24-Solar-terms", "week", "lunar Calendar" and "month" can all improve the models.

To sort the methods of automatically constructing features, the first is the one-hot encoding processing because of its high accuracy and robust generalization. The second is the binding encoding processing. The third is the cross-combination encoding processing. The worst is the polynomial encoding processing.

Through experiments, the key features in the model can often be improved more after processing. The "week" of the one-hot encoding treatment, the "outdoor light intensity" of binning encoding treatment, and the "week" and "carbon dioxide concentration" of polynomial encoding treatment all conform to this rule. Therefore, it is important to find the key original features before processing.

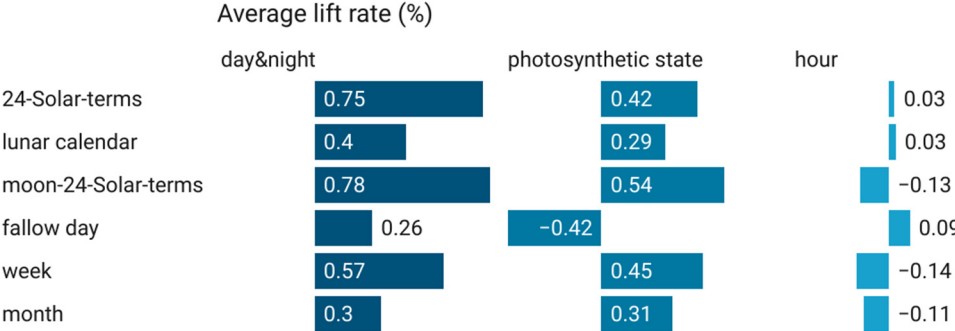

**Figure 15.** Average increase in the $R^2$ of the model (polynomial encoding).

## 4. Conclusions

For high-yield and high-quality horticulture products, the environment construction of CEA buildings is becoming more sophisticated. This leads to higher energy consumption than traditional CEA buildings. It not only leads to high economic and environmental costs, but also may lead to uncertain fluctuations in the power grid. Therefore, energy load prediction for CEA buildings is necessary for energy management. The research on the CEA building energy load model is still few. In this subject, there is less research on features in data-driven methods. This study mainly fills in the research related to the characteristic engineering of CEA building energy consumption prediction model. This technology can help improve the performance of CEA's building energy consumption prediction model. This work has also proved that the time series features and logical features extracted from CEA scenes have significantly improved the performance of the model. Therefore, in practical projects, the energy consumption prediction model of CEA buildings can be developed based on the effective features extracted in this study. The main conclusions of the article are as follows:

Among the eight algorithms, RF has the highest accuracy and the most robust generalization. Under the basic model, the $R^2$ of RF is 0.84~0.94. The variance of RF is 0.002635.

In CEA buildings, "outdoor light intensity", "outdoor temperature", and "week" are the key features for electric load and heat load prediction models. "Carbon dioxide concentration" is a key feature of the power load model. "Indoor humidity" is a key feature of the thermal load model. In addition, some features based on local calendar tools and horticultural production can also significantly improve the performance of the model.

The key feature processing improves the model performance more obviously. In addition, the four treatments' performance is different. The one-hot encoding can improve most models by 1.6%~2.0%. In the binning encoding processing, only the treatment of "outdoor light intensity" can improve the performance of the model. The equal frequency binning encoding and the K-means binning encoding can be increased by 2.2% and 5.0% at most. The cross-combination encoding and the polynomial encoding processing are the worst performers. In addition, their improvement effect does not exceed 1%.

This paper mainly discusses the impact of creating derived features from automatic feature construction and extracting features based on CEA building scenarios on model performance. Compared with the test situation, the actual situation is more complex. The test has some limitations:

1. CEA buildings are often large, and the sensor at a single site sometimes cannot represent the situation of the whole building, especially the temperature, humidity, and light intensity.

2. The random forest algorithm used in the experiment will have different importance of possible features under different algorithm conditions.

3. The test did not consider the proximate characteristics, so the accuracy needs to be further improved.

In the future, to further improve the accuracy of the model, we can use plant images and remote sensing technology to expand feature sources and develop complex algorithms, such as LSTM algorithm, to improve the accuracy of the model.

**Author Contributions:** Conceptualization, Y.C. (Yifan Cao) and X.S.; methodology, Y.C. (Yifan Cao); software, Y.C. (Yifan Cao) and Y.C. (Yangda Chen); data curation, M.S. and C.L.; investigation, W.W., Y.L. and X.G. supervision, X.S.; writing—original draft, Y.C. (Yifan Cao) and Y.C. (Yangda Chen); writing—review and editing, Y.C. (Yangda Chen) and X.S.; project administration, X.S.; unding acquisition, X.S. All authors have read and agreed to the published version of the manuscript.

**Funding:** This research was funded by National Technical System of Bulk Vegetable Industry (CN) grant number CARS-23-C-05, Shaanxi Provincial Technology Innovation Guidance Special Fund (CN) grant number 2021QFY08-02, and Shaanxi Science and Technology Innovation Team (CN) grant number 2021TD-34.

**Data Availability Statement:** The data used to support the findings of this study are available from the corresponding author upon reasonable request.

**Conflicts of Interest:** The authors declare no conflict of interest.

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
