# Peer review of "Impact of Derived Features from the Controlled Environment Agriculture Scenarios on Energy Consumption Prediction Model"

_buildings, doi:10.3390/buildings13010250_

Round 1

Reviewer 1 Report

The abstract part could be improved, by describing an overview of the phenomenon, the gap that the research aims to fill, the research questions or propositions and a brief description of the study contribution.

In the introduction, the focus is on the descriptions of technical features but not on the relevance of the phenomenon, object of the study. For what concern the research questions or propositions, there is a lack of a clear indication.

Moreover there is no section dedicated to the literature review, the theoretical background and the research gap.

Also, there is no description of the theoretical and practical contributions/implications of the study or any description for future research perspectives and limitations.

The research questions or hypotheses are absent. To what question/hypothesis is the study responding?

Lastly, for what concern the development part, the methodology and results are well represented. 

Reviewer 2 Report

Congratulations to the authors for their work.

The work presented is innovative and in my opinion of high quality.

There are some small formatting issues that need to be improved:

 Abstract

The abstract is confused. It should be rewritten so that the reader understands the purpose and methodology of the study, as well as the main results and conclusions.

Introduction

Excessive use of acronyms that make reading difficult

Figures and tables

Figures and tables should be centred on the pages and with homogeneity of typeface and font size. For example, figure 5 has a huge text and is off-centre. In general, all figures and tables should be checked for sharpness, text size and position.

Reviewer 3 Report

Paper minor review.
English should be revised.
Introduction should be improved to present a structured framework, with flowchart to have better understanding of the proposed approach and link to background literature. Link different sections of the paper to the overall framework.
Some figures have no proper legend, with units, and headers for legends.
Some figures are not clear, change color.
English should be revised throughout the paper, first sentence in conclusion has repeated words.
The main contribution is not clear as compared with other studies and key results, this should be further clarified

Round 2

Reviewer 1 Report

Please provide with a section of literature review (what other scholars have studied in relation to the phenomenon object of your study). In this way you can identify and explain the research gap (what are you contributing to the literature). You have written in the conclusions that: " the research is still few...) so if any please mention. Some text editing also need to be reviewed. Thank you for the patience.
